# Selectivity of Terpyridine Platinum Anticancer Drugs for G-quadruplex DNA

**DOI:** 10.3390/molecules24030404

**Published:** 2019-01-23

**Authors:** Elodie Morel, Claire Beauvineau, Delphine Naud-Martin, Corinne Landras-Guetta, Daniela Verga, Deepanjan Ghosh, Sylvain Achelle, Florence Mahuteau-Betzer, Sophie Bombard, Marie-Paule Teulade-Fichou

**Affiliations:** 1Institut Curie, PSL Research University, CNRS-UMR 9187, INSERM U1196, F-91405 Orsay, France; emc.morel@gmail.com (E.M.); Claire.Beauvineau@curie.fr (C.B.); Delphine.Naud@curie.fr (D.N.-M.); Corinne.guetta@curie.fr (C.L.-G.); Daniela.Verga@curie.fr (D.V.); Deepanjan.ghosh@curie.fr (D.G.); sylvain.achelle@univ-rennes1.fr (S.A.); Florence.Mahuteau@curie.fr (F.M.-B.); 2Université Paris Sud, Université Paris-Saclay, CNRS-UMR 9187, INSERM U1196, F-91405 Orsay, France; 3University Rennes, CNRS, ISCR-UMR 6226, F-35000 Rennes, France

**Keywords:** terpyridine platinum complexes, G-quadruplex

## Abstract

Guanine-rich DNA can form four-stranded structures called G-quadruplexes (G4s) that can regulate many biological processes. Metal complexes have shown high affinity and selectivity toward the quadruplex structure. Here, we report the comparison of a panel of platinum (II) complexes for quadruplex DNA selective recognition by exploring the aromatic core around terpyridine derivatives. Their affinity and selectivity towards G4 structures of various topologies have been evaluated by FRET-melting (Fluorescence Resonance Energy Transfert-melting) and Fluorescent Intercalator Displacement (FID) assays, the latter performed by using three different fluorescent probes (Thiazole Orange (TO), TO-PRO-3, and PhenDV). Their ability to bind covalently to the c-myc G4 structure in vitro and their cytotoxicity potential in two ovarian cancerous cell lines were established. Our results show that the aromatic surface of the metallic ligands governs, in vitro, their affinity, their selectivity for the G4 over the duplex structures, and platination efficiency. However, the structural modifications do not allow significant discrimination among the different G4 topologies. Moreover, all compounds were tested on ovarian cancer cell lines and normal cell lines and were all able to overcome cisplatin resistance highlighting their interest as new anticancer drugs.

## 1. Introduction

Among the clinically relevant anticancer drugs, cisplatin is the most frequently used chemotherapeutic drug particularly employed for the treatment of testicular, ovarian, lung, head, and neck cancers. However, undesirable side effects, as well as the emergence of intrinsic or acquired resistances, currently limit its use and drive the design and development of more effective and less toxic analogues [1,2,3]. One of the considered strategies is to induce different DNA lesions and to target specific DNA structures as G-quadruplexes (G4) [4]. G4 consists in the stacking of G-quartets formed by four guanines linked together by reverse-Hoogsteen hydrogen bonds involving their N7. G4s arise from DNA (or RNA) sequences that contain at least four runs of guanines and are stabilized by physiological concentrations of alkali metal cations. A large amount of data provides evidence that such structures could form in cells and may play important roles in biology, such as genomic stability, replication, transcription, translation, and telomere maintenance [5,6]. Bioinformatics [7,8], cellular imaging [9,10,11,12,13], as well as high throughput sequencings of genomic DNA or RNA [14,15,16] have contributed to identifying G4 prevalence at specific key genomic sequences such as telomere and promoter regions of oncogenes. So far, they have been identified as potential drug targets, especially in cancer [17,18,19]. Consequently, a large number of G4 binders based on organic planar structures [20] have been synthesized, including metal complexes [21,22,23,24,25,26,27,28,29]. As an example, RHPS4 was one of the first proof-of-concept drugs for the application of G-quadruplex-binding DNA molecules in anticancer therapies [30]. Among metal complexes, platinum complexes have been extensively studied (for review see References [23,24]). Moreover, platinum (II) can also form coordinate bonds with DNA, and very stable adducts may result [3]. Besides double-stranded DNA, numerous examples indicate that G4 can be covalently coordinated in vitro by platinum complexes either on adenines (N1 or N7) located within the loops or on the guanines (N7) belonging to external G-quartets that transiently open [31,32,33,34,35,36]. Moreover, some of them were shown to target telomeres in cellulo by inducing their uncapping [33], loss [37], dysfunction [38], and platination [39] while other complexes were shown to target c-myc oncogene [40,41]. Despite the large body of data, the mechanism of action and the determinants that drive the in vitro and cellular target recognitions of G4-platinum complexes are not yet established. Metal-terpyridine complexes were shown to have good affinity towards G4 structures, due to their square planar and square-based pyramidal geometries [21,29,36]: it is assumed that the metal ion increases the ability of the ligand to display π–π stacking interactions with the external G-tetrad and can replace a metal cation involved in the G-tetrad stabilization. The family of terpyridine platinum complexes offers the opportunity to analyze in detail a potential structure–activity relationship. Some of them have already been shown to stabilize and metallate in vitro human telomeric and c-myc G4s via selective platination of adenine residues on the loops [31,42,43]. An NMR structure of **Pt-ttpy** complexed with a G4 originating from the promoter region of c-myc oncogene suggests that the predominant interaction occurs through the stacking of **Pt-ttpy** to the outer G-quartet and drives the platination of the adenine residue at the 5’-end overhanging region [42]. All these data suggest that this ligand is well suited for both G4 recognition and metal coordination. In addition, it is important to underline that **Pt-ttpy** has been shown to be able to modify the structure of telomeres of cancerous cell lines by inducing their loss and direct metalation [37,39]. Moreover, one of its derivatives, **Pt-ctpy**, exhibits promising radiosensitization properties in human glioblastoma and lung cancer cells [44]. In conclusion, the terpyridine platinum complexes are, therefore, promising compounds employed for cancer treatment alone or in combination with radiation.

These encouraging properties prompted us to explore the aromatic core around the Pt-terpyridine motif to raise new properties in terms of G4 affinity and selectivity for drug discovery in comparison with **Pt-ttpy** that has already been evaluated on the human telomeric and c-myc G4s [29,31,36,43]. The binding properties of these complexes towards G4s were studied with a panel of oligonucleotides (G4 of various topologies and duplex DNA) using FRET-melting assay and Fluorescent Intercalator Displacement (FID), the latter being performed in the presence of three different probes, namely Thiazole Orange (TO) [45], TO-PRO-3 [46], and PhenDV [47], to determine the key elements for G4 affinity and selectivity. In parallel, platination selectivity and efficiency in competition conditions with duplex DNA have been studied. At last, the potential of these Pt(II)-complexes as antitumor agents has been evaluated by studying their effects on the growth of cisplatin-sensitive and cisplatin-resistant cell lines.

## 2. Results

### 2.1. Panel of Platinum Complexes

Terpyridine (**tpy**) scaffold is a well-used metal ligand and has been extensively derivatized by Vilar et al. to stabilize G4 [22] yet shows poor selectivity toward duplex DNA. To overcome this selectivity issue, the extension of the aromatic surface of the Pt-complexes was successfully proposed. Both **Pt-BisQ** and **Pt-ttpy**, bearing respectively two quinoline moieties or a tolyl group, display higher affinity and selectivity towards telomeric G4 by limiting duplex DNA intercalation [29,31,36]. Our group synthesized **Pt-ctpy**, based on the **Pt-ttpy** scaffold, by adding a short in situ protonable chain [29]. This chain increases the water solubility and the affinity towards telomeric G4 by adding supplementary electrostatic interactions with the G4-DNA [48]. To assess a structure–activity-relationship study, we extended our panel of platinum complexes (Scheme 1).

We changed the central pyridine of the terpyridine to a pyrimidine obtaining compounds **Pt-cpym** and **Pt-vpym**. The ligand **cpym** was prepared by a three-step synthesis starting from the already described pyrimidine 1 (Scheme 2) [49].

A double Stille coupling of dichloropyrimidine **1** afforded compound **2** in very good yield. Deprotection of the phenol group followed by O-alkylation allowed the formation of **cpym**. Compound **vpym**, bearing a vinyl linker, was prepared as described in the literature [50]. Both **vpym** and **cpym** ligands were platinated using PtCl_2_(DMSO)_2_. In addition, the aromatic surface of the complexes has also been extended on the metal center by replacing the labile ligand (Cl^−^) on **Pt-tpy** by a phenylacetylene group (Appendix A). This complex was obtained by transmetallation of the in situ generated copper (I) phenylacetylide on the **Pt-tpy**. **Pt(PA)-tPy** does not contain any labile ligand and consequently is not able to metallate the DNA bases anymore.

### 2.2. Interaction Measurements

Binding properties towards G4 structures of the synthesized complexes were evaluated by performing biophysical assays, such as FRET-melting and G4-FID assays, in the presence of several G4-forming oligonucleotides representative of different folding topologies: 22AG human telomeric sequence (polymorphic), 21CTA human telomeric sequence variant (antiparallel), CEB25-WT minisatellite sequence (parallel with a central long propeller loop), and c-myc protooncogene sequence (parallel with short propeller loops).

The affinity and selectivity for G4-DNA of our panel of platinum complexes have been evaluated by FRET-melting [51] and G4-FID assays [45].

The ligand-induced stabilization measured by FRET-melting experiments (∆*T*_m_) is plotted for all compounds in Figure 1A–E, and the benchmark compound PhenDC3 was used as the control compound. Most importantly, **Pt-BisQ**, **Pt-ctpy**, and **Pt-ttpy** emerge as the best G4 stabilizers of the series, with ∆*T*_m_ values ranging from 10 to 25 °C for all G4-DNA structures (Figure 1A and Appendix A). **Pt-vpym**, **Pt-cpym**, and **Pt-tpy** exhibit lower stabilizing capacities towards G4s while **Pt(PA)-tpy** shows a complete lack of stabilization properties. Among the different topologies exhibiting a similar melting temperature (around 60 °C), all complexes show similar stabilizing properties for 22AG, CEB25-WT, and 21CTA, as compared to c-myc that tends to be less stabilized by the different complexes. In addition, the selectivity of the compounds towards G4s vs. duplex DNA has been evaluated by carrying out competition experiments in the presence of 10 equivalents of ds26 (Figure 1B–E): all complexes show moderate selectivity since their ∆*T*_m_ is partially affected by the presence of ds26. One exception is represented by **Pt-vpym** which has stabilizing properties towards 21CTA and c-myc that increased in the presence of ds26 for a reason that is not yet elucidated but that is likely an artefact.

Since all these complexes (except for **Pt(PA)-tpy**) are susceptible to induce platination reactions, they can be exacerbated by a high temperature and by an increased number of nucleophilic sites which are exposed during unfolding (increased N7 G free sites); we wonder if such a reaction could occur during FRET-melting experiments and to what extent. Therefore, we followed the extent of platination reactions of c-myc and 22AG (same sequences as those used for FRET-melting but without fluorophore labeling) as a function of the melting temperature using ^32^P-labelled oligonucleotides and denaturing gel electrophoresis (Figure 2). **Pt-ttpy**, **Pt-tpy**, and **Pt-BisQ** clearly platinate c-myc and 22AG (except for **Pt-BisQ**) during melting temperature experiments and the extent of platination depends both on the sequence (c-myc (60%) > 22AG (20%)) and on the complex (**Pt-ttpy** > **Pt-tpy** > **Pt-BisQ**). These results suggest that these platination reactions may therefore bias the melting temperature value. **Pt-ctpy**, **Pt-vpym**, and **Pt-cpym** do not give any defined platination product, and thus, they were not evaluated in this melting temperature condition (see Section 2.3. Quadruplex Platination).

Consequently, the relative binding ability of the complexes for the various G4s should be determined in the conditions that are less favorable to platination reaction, the room temperature and short incubation time: all these conditions are gathered in a G4-FID assay. A G4-FID assay is generally performed in the presence of thiazole orange (TO) as an on–off fluorescent probe for DNA structures [48]. TO-G4-FID allowed us to evaluate the binding properties for all the platinum complexes (Figure 3A, Appendix A), except **Pt-vpym** which had spectral properties that are incompatible with TO (Appendix A). The metallic complexes can be ranked in two groups: the first group, including **Pt-BisQ**, **Pt-ctpy**, and **Pt-ttpy**, efficiently displaces TO from all the G4 (>80% displacement at 1 µM) as compared to the duplex ds26 (<20% displacement) (Appendix A), whereas the second group, including **Pt-cpym**, **Pt-tpy**, and **Pt(PA)-tpy**, displaces less efficiently TO (<60%). For all of them, only a small displacement of the probe is observed in the presence of duplex ds26 (Appendix A). As the binding constants of TO for all the tested G4s are in the same range (~10^6^ M^−1^), we could assess that none of the metal complexes has a clear preference for a G4 structure.

However, in order to evaluate **Pt-vpym**, we used the TO-PRO-3-G4-FID assay developed in our group [48] to compare the entire panel of ligands. Of interest, TO and TO-PRO-3 exhibit similar affinity constants towards all the tested G4s structures (*Ka* ~10^6^ M^−1^) [48]. The TO-PRO-3-G4-FID assay shows that **Pt-vpym** displays a moderate affinity for G4 structures (30% displacement) with poor selectivity vs. duplex DNA (Figure 3B, Appendix A) in contrast to **Pt-BisQ**, **Pt-ctpy**, and **Pt-ttpy** (>60% displacement). Of note, only a weak displacement for **Pt-tpy**, **Pt-cpym**, and **Pt(PA)-tpy** has been observed.

In addition, we tested our complexes in a G4-FID assay using PhenDV which is an off–on probe developed in our laboratory [47]. This probe improves the sensitivity of the G4-FID assay, as the read out, different from the two previous reported FID assays, relies on increased fluorescence: PhenDV fluorescence is strongly quenched when it is bound to G4 DNA and fully restored when it is displaced by the ligand. Different from TO and TO-PRO-3, PhenDV displays a higher affinity towards G4 [47] (Table 1) and does not interact with duplex DNA. Thus, the displacement assays were carried on only with G4 sequences. In contrast to TO and TO-PRO-3, the binding constants of PhenDV are largely spread from 10^6^ M^−1^ to 6 × 10^7^ M^−1^ (Table 1). For 21CTA and 22AG, for which PhenDV displays the same affinity, the dye displacement is very similar for each metal complex. At the opposite, as PhenDV is much more affine than TO for CEB25-WT (10^7^ M^−1^ versus 10^6^ M^−1^) and for c-myc (6 × 10^7^ M^−1^ versus 5 × 10^6^ M^−1^), the dye displacement by the complexes are less efficient, especially in the case of c-myc. However, as for TO and TO-PRO-3, the same complex ranking has been observed: **Pt-ctpy**, **Pt-BisQ**, and **Pt-ttpy** are the most affine ligands, and the least affine are **Pt-cpym**, **Pt-tpy**, and **Pt(PA)-tpy** (Figure 3C and Appendix A). In conclusion, FID experiments led to more similar rankings than the FRET experiments for the evaluated metal complexes on the different tested G4.

### 2.3. Quadruplex Platination

Previous studies led by Bertrand et al. have shown that the terpyridine platinum complexes **Pt-tpy** and **Pt-ttpy** were also able to react with the human telomeric G4 (22AG) exclusively with adenines located within the loops whereas **Pt-BisQ** was not able to metallate 22AG [31]. As well, **Pt-ttpy** was found to platinate at the 3’-end exclusively in the proximity of the external G-quartet of 22AG; meanwhile, **Pt-tpy** metallates the most accessible nucleophilic base of 22AG [35], suggesting no tetrad interaction. It has been hypothesized that the **Pt-ttpy** coordination to DNA is, therefore, being driven by its affinity for the G4 structure, a hypothesis that has been supported by the demonstration of the stacking of **Pt-ttpy** on the external quartet of the c-myc derived G4 [42].

The platination reaction of the c-myc oncogene in the presence of our set of candidates (**Pt-tpy**, **Pt-ttpy**, **Pt-BisQ**, **Pt-ctpy**, **Pt-vpym**, and **Pt-cpym**) bearing a labile ligand was followed by gel electrophoresis and the binding sites identified by 3’-exonuclease digestion experiments.

The five platinum complexes were incubated for 18 h at 32 °C with pre-folded 5’-end ^32^P radiolabeled c-myc DNA (10 or 100 µM) and loaded on denaturing polyacrylamide gel electrophoresis (Figure 4).

Two main accelerated bands (A1 and A2) were detected for **Pt-ttpy** and three were detected for **Pt-tpy** (B1, B2, and B3) in the presence of 10 µM of DNA. However, for **Pt-BisQ**, two accelerated bands (C1 and C2) and one retarded band (C3) were only detected at higher DNA concentration (100 µM). It is noteworthy that c-myc platination products migrate faster that the non-platinated G4, resulting in a stark contrast if compared to 22AG in which platination products migrate slower (Figure 2): this is due to the presence of the still folded G4 structures that resist to denaturation, as already found [42]. Non-defined platination products were detected for **Pt-ctpy**, **Pt-cpym**, and **Pt-vpym** on gel electrophoresis at 100 µM of DNA (Appendix A), suggesting that the structure unfolds during platination giving rise to many platinated products. The platination sites of **Pt-ttpy**, **Pt-tpy**, and **Pt-BisQ** were determined by 3’-exonuclease digestion, which stops at the platinated base, followed by a de-platination treatment of the digested fragments with NaCN (Appendix A). The exact length of the digested de-platinated fragment was deduced from its migration compared with the one of a digestion ladder of the c-myc sequence on a denaturating gel electrophoresis.

**Pt-ttpy** forms a platination adduct on the 3’-end of the oligonucleotide, mainly on A21/A22 bases but also, to a lesser extent, on the inner loop of the quadruplex structure, on A12/G13 bases (Figure 5A). In addition, the less affine complex **Pt-tpy** can also form adducts on both flanking sequences of the G4 structure, on T20/A21 at the 3’-end and on G2/A3 on the 5’-end, while the larger aromatic complex **Pt-BisQ** forms exclusively a 3’-end adduct on G19/T20 bases with c-myc (Figure 5B).

### 2.4. Kinetics and Selectivity Studies

The selectivity of c-myc platination produced by **Pt-ttpy** and **Pt-tpy**, the two complexes able to produce efficient platination products at low G4 concentration, was finally assessed by gel electrophoresis by employing the same concentrations used for the FRET-melting experiments. The formation of the platinated products was followed as a function of time (Appendix A) on two ^32^P-radiolabeled DNA, c-myc* and ds26*. **Pt-ttpy** is able to metallate a large amount of G4 DNA c-myc* within 120 min of incubation (Figure 6).

When adding an excess of duplex DNA (ds26) as a competitor, neither the amount of platinated products nor platination kinetics are affected. In contrast, when c-myc was added as a competitor, the platination kinetic of ds26* was affected. These results confirm the high selectivity of the platination reaction performed by **Pt-ttpy** with G4 DNA structures.

In contrast, the platination of c-myc* by **Pt-tpy** is more affected by the presence of the duplex competitor whereas the platination of ds26* is not affected by the addition of c-myc. These results confirm the low binding selectivity of **Pt-tpy** for G4. The amount of G4 adducts obtained with **Pt-tpy** is higher than with **Pt-ttpy**, confirming that **Pt-tpy** can react easily with all the accessible nucleophilic sites of the structure without previous stacking to the external tetrad. In contrast, the reactivity of **Pt-ttpy** is limited to the residues in the vicinity of the external G-quartets which have been shown by NMR to be its main binding sites [42].

### 2.5. In Vitro Cytotoxicity

Finally, we evaluated the effect of the complexes on the growth of two ovarian cancer cell lines A2780 and A2780cis, which are respectively sensitive and resistant to the antitumor drug cisplatin, and one normal lung cell line CCD19Lu. All cell lines have been treated for 96 h with increasing doses of complexes. Platinum complexes show cytotoxicity in the µM range (IC_50_ from 0.08 to 6 µM) as a function of their structure and can be classified as follows: **Pt(PA)-tpy** > **Pt-vpym** > **cisplatin** > **Pt-ttpy** ≥ **Pt-tpy** ≥ **Pt-ctpy** > **Pt-cpym** > **Pt-BisQ** (Table 2). Moreover, none of the platinum complexes show a significant cross-resistance to cisplatin since no significant differences between cisplatin-sensitive and resistant cell lines could be highlighted (IC_50_ratio A2780cis/A2780 < 1.6). However, all of them show no specificity for cancer cell lines, similar to the clinical anticancer drug cisplatin.

## 3. Discussion

In this study, we explored the chemical space around the terpyridine aromatic core to put in evidence the key elements that drive affinity and selectivity for various G4-DNA exhibiting polymorphic (22AG), antiparallel (21CTA), long looped parallel (CEB25-WT), and short looped parallel (c-myc) topologies. The FRET-melting assays pointed out three ligands exhibiting higher binding inducing stabilization independently from the G4 topology (Figure 1A), namely **Pt-BisQ**, **Pt-ctpy**, and **Pt-ttpy** (∆*T*_m_ > 10 °C). Whereas **Pt-vpym**, **Pt-cpym**, and **Pt-tpy** showed less stabilizing capacities (∆*T*_m_ < 10 °C), and no stabilization was observed for **Pt(PA)-tpy**. However, their selectivity for G4 over duplex DNA is moderate since their ∆*T*_m_ decreases significantly in the presence of competitor duplex DNA (Figure 1B–E). This trend was then confirmed by FID assays. Interestingly, PhenDV was a better probe than TO-PRO-3 and TO able to discriminate the relative affinity of the complexes for each G4 topology. Indeed PhenDV, which displays higher binding constants for all of the evaluated G4s, was shown to discriminate more significantly among high affinity G4 ligands [47]. None of the complexes was able to displace PhenDV from c-myc in contrast to TO and TO-PRO-3, the exception represented by the benchmark l ligand, PhenDC_3_. This is consistent with the particularly high affinity of PhenDV for the c-myc structure (Table 1) which induces a harsh competition as compared to the other markers, thereby leading to the selection of only very high affinity ligands (e.g., PhenDC3). Alternatively, other binding sites cannot be excluded for the platinum complexes in the presence of PhenDV.

It is quite surprising that the relative binding properties of the complexes determined from FRET-melting and FID experiments are consistent despite the formation of non-negligible amounts of platinum adducts (up to 50%) during the FRET-meting experiments (Figure 2). This suggests that the platination of the bases already accessible within the loops or released from the G-quartets during the thermal denaturation conditions does not shift significantly the equilibrium towards the unfolded state in FRET conditions. Nevertheless, it could explain the lower ∆*T*_m_ observed on c-myc versus 22AG independently from the metal complex since the amount of platination products is more important for c-myc than 22AG in these conditions. Of note, these FRET-melting experiments, done in thermal-denaturing conditions, lead to an irreversible process. Indeed, the N7 platination of guanines would prevent the formation of the G-quartets and consequently the folding in G4 [52] in thermal renaturation conditions.

Altogether the data highlight the need of different and complementary methods for the determination of the relative affinity of ligands for G4.

Considering the structure–activity relationship, our data show that the extension of the aromatic core modifies the affinity of the complexes for the G4-DNA structures. Since **Pt(PA)-tpy** did not show any affinity for G4, it can be claimed that introducing a phenylacetylene group on the platinum may support a decrease in affinity for the G4. While extending the terpyridine by a tolyl (**Pt-ttpy**), a tolyl with a protonable side chain (**Pt-ctpy**), or replacing the terpyridine core by a bisquinoline (**Pt-BisQ**) increase the affinity of all the complexes for G4-DNA, modifying the terpyridine core (**Pt-vpym** and **Pt-cpym**) reduces the affinity for G4. Interestingly, **Pt-ttpy** showed a high affinity for 22AG. This could be related to the recent results showing that one of the cellular targets of **Pt-ttpy** is indeed telomeric DNA [37,39,44]. Noteworthily, while the selectivity of the present complexes for G4 over duplex DNA is moderate in vitro (Figure 2B–E), the example of **Pt-ttpy** indicates that low selective ligands can reach their cellular target even in the presence of genomic DNA.

One expected property of these complexes is to form mono-adducts with DNA. Our in vitro data show that the relative platination of c-myc G4 over duplex DNA by **Pt-ttpy** and **Pt-tpy** is correlated to the relative affinity for both structures, suggesting that the platination events are driven by the recognition of the DNA structures. However, the relative affinity of present complexes for G4 is not correlated to their platination efficiency. In this regard, **Pt-BisQ**, one of the most relevant complexes, is less susceptible to induce platination of c-myc G4 in vitro if compared to **Pt-ttpy**, **Pt-ctpy**, and **Pt-tpy** and does not show any alkylation on 22AG G4 [31]. This behavior can be explained by i) slow exchange kinetics of the chloride, mandatory for the direct coordination of Pt(II) to the nucleophilic site [53], and ii) the effect produced by the extension of the terpyridine core that can mask the nucleophilic sites of the G4 and/or the accessibility of Pt(II) [42]: indeed, the platinated bases depend on the nature of the complexes (A12, G13, A21, and A22) for **Pt-ttpy** and G19 for **Pt-BisQ**.

Finally and importantly, the IC_50_ of the complexes are in the µM range, except for **Pt(PA)-tpy** which is active at nM concentration in A2780 cell lines. However, all the complexes overcome the cisplatin resistance in A2780cis cell lines, providing new interesting anticancer drug candidates. This suggests that they may enter cells via a pathway independent from the copper carrier proteins used by cisplatin [54]. Among them, **Pt(PA)-tpy** is the most efficient complex despite its inability to form DNA adducts. Therefore, our results pointed out that the cellular efficiency of a panel of platinum complexes is not strictly related to their affinity for the various G4-DNA structures or their capacity to platinate in vitro DNA structures. Many other factors, in addition to their cellular target, must be taken into account, such as their cellular uptake and their binding to genomic DNA, as already shown for other platinum complexes [33,55]. For example, previous studies showed that the genomic DNA binding of **Pt-tpy** and **Pt-ttpy** is less efficient than the one of cisplatin, as compare to their cellular uptake.

In conclusion, the modulation of the terpyridine core of platinum complexes may greatly influence their in vitro affinity for G4 and their capacity to induce specific metallation of G4 in vitro and may be promising as future anticancer drugs overcoming the resistance to cisplatin. The identification of the cellular targets, which is ongoing, could definitely indicate if they also represent potential drugs targeting G4 in cellulo.

## 4. Materials and Methods

### 4.1. Materials

Oligonucleotides purified by reversed-phase HPLC were purchased from Eurogentec (Angers, France). The dual fluorescently labeled oligonucleotides were purchased from Eurogentec. The donor fluorophore was 6-carboxyfluorescein (FAM) and the acceptor fluorophore was 6-carboxytetramethylrhodamine (TAMRA).

PhenDV was synthesized as already described [47].

Stock solutions of the ligands (2 mM in DMSO) were used for G4-FID, FRET-melting assay, and fluorimetric titration, unless otherwise stated, and were stored at −20 °C. TO, TO-PRO-3, and cacodylic acid were purchased from Aldrich and used without further purification. Stock solutions of TO (2 mM in DMSO), PhenDV (2 mM in DMSO), and TO-PRO-3 (1 mM in DMSO) were used for the G4-FID assay. Fluorescent probe powders and solutions were stored and used, protected from light, and used as aliquots to avoid freeze–thaw cycles.

The FRET-melting measurements are performed on a 7900HT Fast Real-Time PCR System (Applied Biosystems, Foster City, CA, USA) with a Microamp Fast optical 96-well reaction plate (Applied Biosystems). HT-G4-FID measurements were performed on a FLUOstar Omega microplate reader (BMG Labtech, Champigny-sur-Marne, France) with 96-well Non-Binding Surface black with black bottom polystyrene microplates (Corning). Fluorescence measurements (i.e., fluorimetric titration) were performed on a Cary Eclipse Fluorescence spectrophotometer (Agilent Technologies, Les Ulis, France).

### 4.2. Organic Synthesis

^1^H and ^13^C spectra were recorded at 300 MHz and 75 MHz on a Bruker Avance 300 spectrometer and at 500 MHz and 126 MHz on a bruker Avance 500 spectrometer (at the NMR service of ICSN) using TMS as the internal standard (Appendix A). DMSO-d_6_ and CDCl_3_ were purchased from SDS. Proton chemical shifts are reported in ppm (δ) with the solvent reference as the internal standard (DMSO-d_6_, δ 2.50 ppm; CDCl_3_, δ 7.26 ppm). Data are reported as follows: chemical shift (multiplicity (singlet (s), doublet (d), triplet (t), and multiplet (m)), coupling constants (Hz), and integration). LC-MS spectra (ESI in the positive ion mode) were performed with a Waters ZQ instrument (source voltage 50–75 kV). High resolution mass spectrometry (HR-MS) was performed at the Small Molecule Mass Spectrometry platform of IMAGIF (Centre de Recherche de Gif, Gif-sur-Yvette, France). Analytical thin-layer chromatography (TLC) was performed using silica gel 60 Å UV254 precoated plates (0.20 mm thickness) from Macherey Nagel (Hoerdt, France). Visualization was accomplished by irradiation with a UV lamp. Preparative flash chromatography was carried on a CombiFlash Companion) from Teledyne Isco(Lincoln, NE, USA) equipped with packed silica cartridges from Interchim (Montluçon, France). Starting materials were purchased from Sigma-Aldrich (Lyon, France), Alfa Aesar (Karlsruhe, Germany), and Acros (Geel, Belgium). The 2,4-dichloro-6-(4-methoxyphenyl)pyrimidine (CAS [154499-86-2]) [49], vpym (CAS [1297529-36-2]) [50] were prepared as described in the literature.

*4-(4-Methoxyphenyl)-2,6-bis(pyridin-2-yl)pyrimidine* (**1**). ^1^H-NMR (300 MHz, CDCl_3_): δ (ppm) 8.91 (d, *J* = 3.0 Hz, 1H), 8.76 (m, 4H), 8.39 (d, *J* = 9.0 Hz, 2H), 7.92 (m, 2H), 7.58–7.38 (t, *J* = 8.0 Hz, 2H), and 7.06 (d, *J* = 9.0 Hz, 2H), 3.91 (s, 3H); ^13^C-NMR (75 MHz, DMSO-d_6_): *δ* (ppm) 165.2, 163.9, 163.6, 162.1, 155.6, 154.4, 150.1, 149.4, 137.1, 136.8, 129.5, 129.2, 125.6, 124.6, 124.1, 122.4, 114.2, 111.1, and 55.4; and LR-MS (ESI-MS) *m*/*z* = 341.13 [M + H]^+^.

In a dry round-bottomed flask, 2,4-dichloro-6-(4-methoxyphenyl)pyrimidine (677 mg, 2.65 mmol, 1.0 eq) and Pd(Ph_3_)_4_ (460 mg, 0,39 mmol, 0.15 eq) are introduced under argon atmosphere. Dry toluene (20 mL) is added, and a cooling system is installed. The mixture is degassed for 10 min before the addition of 2-(tributylstannyl)pyridine (2.1 mL, 6.63 mmol, 2.5 eq) and heated at reflux (110 °C) for 15 h. Then, 30 mL of water is added. The crude mixture is filtrated on a pad of celite and washed with ethyl acetate, followed by extraction. The combined organic phase is dried over MgSO_4_, filtered and concentrated to dryness. The product is purified by column chromatography (Al_2_O_3_—cyclohexane/AcOEt 50/50) to afford the expected product (749 mg, 83%).

*4-(2,6-Bis(pyridin-2-yl)pyrimidin-4-yl)phenol* (**2**). ^1^H-NMR (300 MHz, DMSO-d_6_): *δ* (ppm) 10.15 (s, 1H), 8.84 (s, 2H), 8.74 (s, 1H), 8.65 (t, *J* = 7.5 Hz, 2H), 8.29 (d, *J* = 7.0 Hz, 2H), 8.07 (dt, *J* = 14.0, 7.0 Hz, 2H), 7.76–7.51 (m, 2H), and 6.99 (d, *J* = 9.0 Hz, 2H); ^13^C-NMR (75 MHz, DMSO-d_6_): *δ* (ppm) 165.3, 163.0, 161.6, 156.2, 151.8, 149.5, 147.5, 146.9, 143.4, 138.6, 129.9, 128.9, 126.9, 126.1, 125.7, 122.8, 116.1, and 111.9; and LR-MS (ESI-MS) *m*/*z* = 327.31 [M + H]^+^.

Compound (**1**) (200 mg, 0.58 mmol, 1.0 eq) is dissolved in anhydrous CH_2_Cl_2_ (10 mL) under argon atmosphere. The mixture is cooled down to −78 °C, and BBr_3_ (200 µL, 2.11 mmol, 3.6 eq) is added dropwise. After cooling back to room temperature, the mixture is stirred for one night and quenched by ice addition. The red solid is filtered, washed, and crystallized in methanol (96 mg, 50%).

*2,4-Di(pyridin-2-yl)-6-(4-(2-(pyrrolidin-1-yl)ethoxy)phenyl)pyrimidine* (**cpym**). ^1^H-NMR (300 MHz, CDCl_3_): *δ* (ppm) 8.86 (m, 1H), 8.71 (m, 4H), 8.32 (d, *J* = 9.0 Hz, 2H), 7.85 (m, 2H), 7.37 (m, 2H), 7.02 (d, *J* = 9.0 Hz, 2H), 4.15 (t, *J* = 6.0 Hz, 2H), 2.89 (t, *J* = 6.0 Hz, 2H), 2.60 (m, 4H), and 1.77 (m, 4H); ^13^C-NMR (75 MHz, CDCl_3_): *δ* (ppm) 165.3, 164.0, 163.7, 161.5, 155.7, 154.5, 150.1, 149.5, 137.3, 137.0, 129.6, 129.3, 125.5, 124.8, 124.2, 122.5, 114.9, 111.3, 77.6, 77.2, 76.7, 67.3, 55.1, 54.9, and 23.6; LR-MS (ESI-MS) *m*/*z* = 424 [M + H]^+^; HR-MS (ESI+) *m*/*z* = 424.2137; and found, 424.2125.

In a round-bottomed flask under argon atmosphere, 4-(2,6-bis(pyridin-2-yl)pyrimidin-4-yl)phenol (50 mg, 0.15 mmol, 1.0 eq), N-(2-chloroethyl)pyrrolidine hydrochloride (26 mg, 0.15 mmol, 1.0 eq), and cesium carbonate (149 mg, 0.46 mmol, 3 eq) are introduced with DMF (2 mL). The mixture is stirred overnight at 100 °C. The solvent is removed under vacuum. Ethyl acetate (20 mL) and water (20 mL) are added for extraction. The combined organic phase is dried over MgSO_4_, filtered, and concentrated to dryness to afford the compound **cpym** (58 mg, 89%).

*Chloro-(2,4-di(pyridin-2-yl)-6-(4-(2-(pyrrolidin-1-yl)ethoxy)phenyl)pyrimidine)-platinum(II) Chloride* (**Pt-cpym**). ^1^H-NMR (300 MHz, DMSO-d_6_): *δ* (ppm) 9.25 (s, 1H), 9.04 (d, *J* = 8.0 Hz, 1H), 8.75 (dd, *J* = 10.0, 5.5 Hz, 2H), 8.67–8.40 (m, 5H), 7.96 (dd, *J* = 12, 5.5 Hz, 2H), 7.16 (d, *J* = 9.0 Hz, 2H), 4.31 (t, *J* = 4.5 Hz, 2H), 3.02 (s, 2H), 2.74 (m, 4H), and 1.78 (m, 4H); ^13^C-NMR (126 MHz, DMSO-d_6_): δ 166.2, 163.4, 161.8, 159.5, 156.5, 154.6, 151.8, 151.5, 143.0, 142.7, 131.0, 130.94, 130.7, 127.7, 127.2, 126.5, 115.5, 113.1, 54.1, 53.8, 29.2, and 23.1; LR-MS (ESI-MS) *m*/*z* = 654 [M + H]^+^; HR-MS (ESI+): *m*/*z* = 653.1395 calculated for C_26_H_25_N_5_OClPt; and found, 653.1367.

In a round-bottomed flask under argon atmosphere, cpym (40 mg, 0.09 mmol, 1.0 eq) is dissolved in a minimal amount of CH_2_Cl_2_ (3 mL). Platinum catalyst Pt(DMSO)_2_Cl_2_ (40 mg, 0.09 mmol, 1.0 eq) is introduced dropwise with methanol (3 mL). The mixture is stirred at 50 °C for 20 h. The crude product is filtered on membrane, washed with a mixture of MeOH/CH_2_Cl_2_/acetone, and dried by Et_2_O. **Pt-cpym** is obtained as a dark powder (20 mg, 31%).

*Chloro-(E)-4-(2-(2,6-di(pyridin-2-yl)pyrimidin-4-yl)vinyl)-N,N-dimethylaniline-platinum(II) Chloride* (**Pt-vpym**). ^1^H-NMR (500 MHz, DMSO-d_6_): δ 8.93 (d, *J* = 5.0 Hz, 1H), 8.90 (d, *J* = 5.0 Hz, 1H), 8.65 (d, *J* = 7.5 Hz, 1H), 8.60–8.50 (m, 3H), 8.48 (d, *J* = 7.5 Hz, 1H), 8.24 (d, *J* = 15.5 Hz, 1H), 8.05–7.99 (m, 2H), 7.58 (d, *J* = 9.0 Hz, 2H), 7.05 (d, *J* = 15.5 Hz, 1H), 6.72 (d, *J* = 9.0 Hz, 2H), and 3.04 (s, 6H);^13^C-NMR (126 MHz, DMSO-d_6_): δ 166.6, 161.6, 157.9, 156.8, 155.2, 152.5, 151.8, 151.6, 144.8, 142.8, 142.5, 131.1, 130.7, 130.3, 126.7, 126.6, 122.0, 118.4, 114.2, 112.1, and 45.7; LR-MS (ESI-MS) calculated for [C_24_H_21_N_5_ClPt]^+^, *m*/*z* = 610.22 [M + H]^+^; HR-MS (ESI+): *m*/*z* = 609.1133; and found, 609.1140.

In a dry round-bottomed flask under argon atmosphere, **vpym** (50 mg, 0.13 mmol, 1.0 eq) is dissolved in the minimal amount of CH_2_Cl_2_ (3 mL). Platinum catalyst Pt(DMSO)_2_Cl_2_ (55.9 mg, 0.13 mmol, 1.0 eq) is introduced dropwise with methanol (3 mL). The dark reaction mixture is stirred overnight at room temperature. The crude product is filtrated on nylon membrane, washed with a mixture of MeOH/CH_2_Cl_2_/acetone, and dried by Et_2_O. **Pt-vpym** is obtained as a dark purple solid (54 mg, 74%).

*Phenylethylny-(2,6-bis(pyridin-2-yl)pyridine)-platinum(II) Hexafluorophosphate* (**Pt(PA)-tpy**). ^1^H-NMR (300 MHz, DMSO-d_6_): δ 9.05 (d, *J* = 4.8 Hz, 1H), 8.77–8.31 (m, 6H), 8.01–7.73 (m, 2H), 7.48 (d, *J* = 7.1 Hz, 2H), and 7.44–7.08 (m, 5H); and LR-MS (ESI+): *m*/*z* = 529.9 [M + H]^+^.

**Pt-tpy** (100 mg, 0.216 mmol), phenylacetylene (47.5 µL, 0.431 mmol), copper iodide (4.11 mg, 0.022 mmol), and triethylamine (30.0 µL, 0.216 mmol) were dissolved in DMF (10 mL) to give a bright red suspension. The reaction was stirred with a magnetic stir bar at room temperature under argon for 3 days. The solution slowly turned dark green. An aqueous saturated solution of NH_4_PF_6_ was added. The dark green precipitate was filtered and washed with water and plenty of diethyl ether (yellow filtrate) until the filtrate turned black and was finally dried to afford a green powder (42.3 mg, 0.080 mmol, 37.1% yield).

### 4.3. Oligonucleotides

For FI, fluorimetric, and gel electrophoresis experiments,
22AG5’-A GGGTTAGGGTTAGGGTTAGGG-3’c-myc (myc22)5’-TGAGGGTGGGTAGGGTGGGTAA-3’ds265’-CAATCGGATCGAATTCGATCCGATTG-3’21CTAAGGGCTAGGGCTAGGGCTAGGGCEB25-WTAAGGGTGGGTGTAAGTGTGGGTGGGT

For FRET experiments,
F-21-T5’-*FAM*-GGG TTA GGG TTA GGG TTA GGG-*TAMRA*-3’F-myc-T5’-*FAM*-TGA GGG T GGG TA GGG T GGG TAA-*TAMRA*-3’F-21CTA-T5’-*FAM*-AGGGCTAGGGCTAGGGCTAGGG- *TAMRA*-3’F-CEB25-WT-T5’-*FAM*-AGGGTGGGTGTAAGTGTGGGTGGGT- *TAMRA*-3’

### 4.4. Preparation of Oligonucleotides

For the G4-FID assay, the oligonucleotides were dissolved in K^+^100 buffer (10 mM lithium cacodylate buffer pH 7.3, 100 mM KCl, 1% DMSO). Oligonucleotide concentrations were determined on the basis of their absorbance at 260 nm. For the fluorimetric titration, the oligonucleotides were dissolved in K^+^100 buffer without DMSO. For the FRET-melting assay, the oligonucleotides were dissolved in K^+^1 buffer (10 mM lithium cacodylate buffer pH 7.3, 1 mM KCl, 99 mM LiCl) except for F-21-T. F-21-T was dissolved in K^+^10 buffer (10 mM lithium cacodylate buffer pH 7.3, 10 mM KCl, 90 mM LiCl).

Prior to use, all oligonucleotides were pretreated by heating at 95 °C for 5 min, then rapidly cooled to 4 °C to favor the intramolecular folding by kinetic trapping. Duplex-DNA ds26 was prepared by heating the self-complementary strand at 90 °C for 5 min in K^+^1 buffer followed by a slow cooling over 6 h.

### 4.5. FRET-Melting Experiments

The stabilization of the compounds with a quadruplex-structure was monitored via FRET-melting assay performed in 96-well plates on a real-time PCR apparatus 7900HT Fast Real-Time PCR System as follows: 5 min at 25 °C and then an increase of 0.5 °C every minute until 95 °C. Each experimental condition was tested and replicated in a volume of 25 μL for each sample. The FRET-melting assay was performed with three dual fluorescently labeled DNA oligonucleotide sequences oligonucleotides. The donor fluorophore was 6-carboxyfluorescein, FAM, and the acceptor fluorophore was 6-carboxytetramethylrhodamine, TAMRA. The 96-well plates (Applied Biosystems) were prepared by aliquoting the annealed DNA (0.2 μM in K^+^1 or K^+^10 buffer) into each well, followed by 1 μL of the ligand (100 μM (5 eq) in DMSO). For competition experiments, duplex competitors were added to 200 nM quadruplex sequences at final concentrations of 3.0 μM (15 eq) and 10.0 μM (50 eq), with a total reaction volume of 25 μL, with the labeled oligonucleotide (0.2 μM) and the ligand (1 μM). Measurements were made with excitation at 492 nm and detection at 516 nm. The change in the melting temperature at 1.0 μm compound concentration, ΔTm (1.0 μM), was calculated from at least two experiments by subtraction of the blank from the averaged melting temperature of each compound (1.0 μM). The final analysis of the data was carried out using Origin Pro 8.6 data analysis.

### 4.6. HT-G4-FID Assay

Each G4-FID assay was performed in a 96-well Non-Binding Surface black with black bottom polystyrene microplates (Corning). Every ligand was tested on a line of the microplate, in duplicate. The microplate was filled with (a) K^+^100 solution (qs for 200 μL), (b) 10 μL of a solution of pre-folded oligonucleotides (5 μM) and fluorescent probe (TO/TO-PRO-3 (10 μM—2 eq) or PhenDV (7,5 µM—1.5 eq)), and (c) an extemporaneously prepared 5 μM ligand solution in K^+^100 buffer (0 to 100 μL) along the line of the microplate, i.e., from column A to column H: 0, 0.125, 0.25, 0.375, 0.5, 0.625, 0.75, 1.0, 1.25, 1.5, 2.0, and 2.5 μM. After 5 min of orbital shaking at 500 rpm, fluorescence is measured using the following experimental parameters: positioning delay of 0.5 s, 20 flashes per well, emission/excitation filters for TO at 485/520, TO-PRO-3 at 620/670, and PhenDV at 355/520 gain adjusted at 80% of the fluorescence from the most fluorescent well (i.e., a well from column A for TO and a well from column H).

The percentage of TO displacement is calculated from the fluorescence intensity (F), using
(1)$$\%\;TO\;displacement=1-\frac{F}{F_{0}}$$
where F_0_ is the fluorescence from the fluorescent probe bound to DNA without added ligand.

In the case of PhenDV, the fluorescence of the unbound probe is not negligible. The percentage of displacement becomes
(2)$$\%\;PhenDV\;Displacement=\frac{F-F_{0}}{F_{ligand+probe}-F_{0}}$$

The PhenDV displacement is calculated from the fluorescence intensity F; F_(ligand+probe)_, which refers to the fluorescence of the probe in presence of the ligand (without G4); and F_0_, which is the fluorescence without added ligand. The term F_(ligand+probe)_ is necessary as the ligand can quench the fluorescence of the probe. The percentage of displacement is then plotted as a function of the concentration of the added ligand. The DNA affinity was evaluated by the concentration of ligand required to decrease the fluorescence of the probe by 50%, was noted DC50, and was determined after non-linear fitting of the displacement curve.

### 4.7. Fluorimetric Titrations for Affinity Constant Evaluation Reported in Table 1

A temperature of 20 °C was kept constant with a thermostated cell holder. Each titration was performed in a 1 mL quartz cell in K^+^100-buffer in a total volume of 1 mL. Titrations were performed with a solution of the fluorescent probe (TO; 0.5 µM or PhenDV; 1 μM) in the corresponding buffer in which gradual addition of oligonucleotides was carried out (up to 10 molar equivalents). After each addition, a fluorescence emission spectrum was recorded at 501 or 387 nm excitation wavelength, respectively. The fluorescence emission area was measured between 510–750 or 397–700 nm, respectively, with 1.0 nm increments, a 0.1 s integration time, and 3/3 nm (excitation/emission) slits. The titration curves were obtained by plotting the fluorescence emission area enhancement against the oligonucleotide concentration. Fluorimetric titrations were performed according to published procedures [56], and the binding constants were determined by fitting of the experimental data to the theoretical model:(3)$$\frac{I}{I_{0}}=1+\frac{Q-1}{2}\left(A+xn+1-\sqrt{{\left(A+xn+1\right)}^{2}-4xn}\right)$$
where *Q* = *I*_∞_/*I*_0_ is the minimal fluorescence intensity in the presence of excess ligand; *n* is the number of independent binding sites per quadruplex; *A* = 1/(K_b_ × *c*_L_); and *x* = *c*_G4_/*c*_L_ is the titration variable.

### 4.8. Gel Electrophoresis

The oligonucleotides were 5’-end-labelled using a polynucleotide kinase and (γ^32^P)-ATP (Perkin Helmer). The reaction products were purified by electrophoresis on 20% denaturing gel. Sample platinations were prepared by folding a mixture of 5’-end-radiolabeled DNA and 10 µM or 100 µM of non-radiolabeled material in 100 mM KClO_4_ or 100 mM KCl solution except for the platination reaction in the denaturing temperature conditions. The folding was achieved by heating the samples at 90 °C for 5 min, followed by slow cooling to room temperature over the course of 2 h to induce the formation of the quadruplex structure. It was then incubated with platinum complexes, and platinated products were separated by electrophoresis on 15% denaturing gel. They were then eluted from gel, precipitated, treated by 3’-exonuclease at 37 °C for 30 min, and loaded on a 20% denaturing gel. The digested fragments were eluted from this gel precipitated, treated over night by NaCN 0.2M, precipitated, and loaded once again on a 20% denaturing gel. Gels were scanned using a STORM860 (Molecular Dynamics).

### 4.9. Cell Culture

The ovarian carcinoma cell lines and human normal lung cells were purchased from ATCC and were grown in complete RPMI medium (ovarian) and DMEM (human lung cells) supplemented with 10% fetal calf serum, in the presence of penicillin, streptomycin. The resistance of A2780cis cells to cisplatin was maintained by monthly treatment with 1 µM cisplatin for 4 days. Concentrated stock solutions of complexes were conserved at −20 °C (DMSO/Water) and freshly diluted in water just before the experiments. Cells were treated with various concentrations of platinum complexes at 37 °C under humidity and 5% CO_2_ conditions for 96 h. Cellular growth was quantified using the particle counter MOXI (VWR).

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
