# Peer review of "Selectivity of Terpyridine Platinum Anticancer Drugs for G-quadruplex DNA"

_molecules, 2019, doi:10.3390/molecules24030404_

Round 1
Reviewer 1 Report
Review is attached.

Author Response
Reviewer 1
This article by Teulade-Fichou et al. describes the synthesis of novel terpyridine-platinum complexes, as well as the evaluation of their binding to G-quadruplex structures carried out by FRET-melting, FID assay and gel electrophoresis. The authors also investigated the in vitro anticancer activity of the analyzed complexes on cisplatin-sensitive or resistant ovarian cancer cell lines.
The manuscript needs major revisions before being published in Molecules. Some formal errors and typos should be removed. Additional experiments are necessary to improve the overall solidity of the data. The figures style should be harmonized.
We thank the Reviewer for his comments and recommendations, and we revised the manuscript accordingly. The modifications are boxed in yellow in the revised manuscript.
Line 122: please explain why you chose F-21-T, F-c-myc-T, F-CEB25-WT-T and F-21CTA-T as model G4-forming oligonucleotides. Which is their biological relevance? Are they taken from regulatory genomic regions? Which kind of G4 topology they adopt?
This has been explained and the paragraph has been moved lines 109-113
Binding properties towards G4 structures of the synthesized complexes were evaluated by performing biophysical assays, such as FRET melting and G4-FID assay, in the presence of several G4-forming oligonucleotides representative of different folding topologies: 22AG human telomeric sequence (polymorphic), 21CTA human telomeric sequence variant (anti-parallel), CEB25WT minisatellite sequence (parallel with a central long propeller loop), and c-myc protooncogene sequence (parallel with short propeller loops).
Figure 1: please adopt the same scale on the Temperature axis for all the panels B-C-D-E; put the complexes in the same order on the vertical axis; use bigger font; align the four panels to allow an easy comparison of the ΔTm on horizontal axis.
This has been done.
Line 140: please clarify that the sequences used in gel experiments are the same used for FRET melting, except for the fluorophore and quencher; explain why the experiments were performed only for c-myc and 22AG, excluding CEB25-WT and 21CTA.
Explanations have been provided in the text and in material and methods.
Therefore, we followed the extent of platination reactions of c-myc and 22AG (same sequences as those used for FRET melting but without fluorophore labeling) as a function of the melting temperature using 32P-labelled oligonucleotides and denaturing gel electrophoresis
Figure 2: please add gel experiments for Pt-ctpy and delete the one of Pt-tpy: in this way, it seems to be a rational choice behind this experiment, namely choosing the three most stabilizing G4 complexes.
As shown in figure S5, no defined platination product was detected for Pt-ctpy: it appeared as a smear on gel electrophoresis impeding the quantification of all the products. This is the reason why this compound was not studied in the platination reaction in melting temperature conditions. Pt-tpy was used as a non-selective binding control compound in figures 2,4 and 6. The following sentence has been added.
Pt-ctpy, Pt-vpym and Pt-cpym do not give any defined platination product thus they were not evaluated in this melting temperature condition (see 2.3 Quadruplex platination).
Line 153: delete ‘Figure 2’.
Line 159: delete ‘Figures S2 and S3’ and write ‘Figures S2 and S4’.
Line 189: delete ‘and S4’.
Line 198: at 25 °C or 32 °C (as written below in the text)?: 32°C
Line 157: delete ‘Figure S2’ and write ‘Figure S3’;
Line 209: there are no data at 10 μM DNA concentration in figure S5, please delete ‘10 μM’.
We thank the Reviewer for pointing out these mistakes, which have been corrected in the revised manuscript
In Figure S3, it could be more appropriate to add spectra for all the investigated complexes and insert the Thiazole Orange spectrum; please explain the incompatibility of Pt-vpym for FID assay or add a reference.
Figure S3 has been modified
Lines 171-174: this sentence is not clear, please rewrite.
The sentence has been rewritten as following
Of interest, TO, TO-PRO-3 exhibit similar affinity constant towards all the tested G4s structures (Ka~106 M-1) [47]. TO-PRO-3-G4-FID assay shows that Pt-vpym displays moderate affinity for G4 structures (30% displacement) with poor selectivity for G4 vs duplex DNA (Figures 3B, S2 and S4), in contrast to Pt-BisQ, Pt-ctpy and Pt-ttpy (>60% displacement). Of note, only a weak displacement for Pt-tpy, Pt-cpym, and Pt(PA)-tpy has been observed.
Line 199: clarify why different concentrations were used for c-myc when incubated with different complexes.
The concentration of DNA used for c-myc platination depends on the reactivity of the Pt-complex (10 and 100µM). Indeed platination of c-myc by Pt-BisQ was not efficient at 10µM and thus this compound was used at 100µM. With the aim to compare the results of all the complexes, the platination reactions done at 100µM are shown in figure S5.
Figure 4: why platinated complexes migrated faster than free DNA? Please explain.
Faster migration of the platinated product is due to the still folded G4 structures that resist to denaturation. Indeed, when heated, the sample migrates slower. This has already been shown and discussed in reference 41. The following sentence has been added.
It is noteworthy that c-myc platination products migrate faster that the non-platinated G4 in contrast to the 22AG, which platination products migrate slower (Figure 2). This is due to the presence of the still folded G4 structures that resist to denaturation, as already found [43]
Figure 5: for a better visualization, the c-myc backbone should be one color and only the platinated residues highlighted in a different color.
The sequence showing the platination sites has been added.
Lines 223-230: These sentences should be shifted at the beginning of the related paragraph: in this way the authors can justify why they performed the experiments only on c-myc sequence.
This has been done
Lines 232-253: please add kinetics and selectivity studies also for Pt-BisQ. Why authors excluded this compound? It is one of the most interesting ones here studied, as stated above in the text.
The kinetics and selectivity studies for Pt-BisQ have not been performed because efficient platination was obtained only at high G4 concentration (100µM instead of 10µM for Pt-ttpy and Pt-tpy). The following sentence has been modified
The selectivity of c-myc platination produces by Pt-ttpy and Pt-tpy, the two complexes able to produce efficient platination products at low G4 concentration, was finally assessed by gel electrophoresis by employing the same concentrations used for FRET-melting experiments
Lines 255-262: please add in vitro data also for normal cells to evaluate if some selectivity emerges for these novel synthesized complexes compared to cisplatin.
The evaluation of the complexes on normal cells has been done and added in table 2.
Moreover, none of the platinum complexes show significant cross-resistance to cisplatin, since no significant differences between cisplatin-sensitive and resistant cell lines could be highlighted (IC50ratio A2780cis/A2780<1.6). However, all of them show no specificity for cancer cell lines, similar to the clinical anti-cancer drug cisplatin.
Line 504: fluorimetric titrations are not described above, please delete this paragraph from ‘Materials and Methods’.
The fluorometric titrations have indeed been done to evaluate the affinity constants Ka of PhenDV and TO for the various G4 (Table 1). For the sake of clarity the title has been modified as follows
4.7. Fluorimetric titrations for affinity constant evaluation reported in Table 1
Figure S1: NMR spectra should be divided in multiple figures, each with its own description.
The figures have been annotated (A-K) for clarity
Figure S2: please remove sequence names from the graphs, use the same scale in y-axis for all the
panels, use bigger font.
The modifications have been done, except for the Y axis that is specific for each experiment.
References: many papers from the authors are cited (e.g. in the part where refs 30-35 are present). Particularly in the Introduction, additional, recent works - from different research groups - on the reactivity and interaction between Pt(II) complexes and G-quadruplex structures, or in general DNA model systems, should be mentioned to provide the readers a more complete view of the current literature (ca. 2015-2018) on this topic.
The references have been updated and completed with two recent reviews. To be highlighted, the purpose of this paper is to focus the attention and, as a consequence the introduction on platinum complexes that were shown to coordinate/platinate G-quadruplex structures, which per se restricts the number of related studies.
Among metal complexes, platinum complexes have been extensively studied (for review see [23, 24]). Moreover, platinum(II) can also form coordinate bonds with DNA and very stable adducts may result [31]. Beside double-stranded DNA, numerous examples indicate that G4 can be covalently coordinated in vitro by platinum complexes either on adenines (N1 or N7) located within the loops or on the guanines (N7) belonging to external G-quartets that transiently open [32-37]
Reviewer 2 Report
The manuscript entitled “Selectivity of Terpyridine platinum anti-cancer drugs for G-quadruplex DNA” explored a panel of terpyridine platinum(II) complexes bound to various of G-quadruplex (G4) topologies. The authors studied the affinity and selectivity of theses platinum complexes towards different G4s. And the platination efficiency were also studied. The author showed how the terpyridine core of platinum complexes affected the selectivity for G4s. The results are interesting and important for designing platinum(II) complexes targeting G4s. This work can be published in the journal Molecules after addressing following issues.
1. Figure 2: Can the author explain why the bands of c-myc platination products shifted faster than the non-platination bands (figure 2A), but the bands of 22AG platination products shifted slower (figure 2B)? And what is the meaning of “T”, which labeled in the first bands of figure 2A and 2B?
2. Please give FRET ΔTm values and G4-FID results in Table.
Author Response
The manuscript entitled “Selectivity of Terpyridine platinum anti-cancer drugsfor G-quadruplex DNA” explored a panel of terpyridine platinum(II) complexes bound to various of G-quadruplex (G4) topologies. The authors studied the affinity and selectivity of theses platinum complexes towards different G4s. And the platination efficiency were also studied. The author showed how the terpyridine core of platinum complexes affected the selectivity for G4s. The results are interesting and important for designing platinum(II) complexes targeting G4s. This work can be published in the journal Molecules after addressing following issues.
We thank the Reviewer for his comments and recommendations, and we revised the manuscript accordingly. The modifications are boxed in yellow in the revised manuscript.
Figure 2: Can the author explain why the bands of c-myc platination products shifted faster than the non-platination bands (figure 2A), but the bands of 22AG platination products shifted slower (figure 2B)?
Faster migration of the platinated product is due to the still folded G4 structures that resist to denaturation. Indeed, when heated, the sample migrates slower. This has already been shown and discussed in reference 41. The following sentence has been added in the paragraph
It is noteworthy that c-myc platination products migrate faster that the non-platinated G4 in contrast to the 22AG, which platination products migrate slower (Figure 2). This is due to the presence of the still folded G4 structures that resist to denaturation, as already found [43]
And what is the meaning of “T”, which labeled in the first bands of figure 2A and 2B? 2. Please give FRET ΔTm values and G4-FID results in Table.
“T” was used for “untreated sample” at time 0. It has been removed and replaced by “0”
FRET ΔTm values and G4-FID results in Tables S1, S2 S3 and S4 in supplementary information
Reviewer 3 Report
Review of the manuscript Molecules 417755 entitled “Selectivity of terpyridine platinum anti-cancer drugs for G-quadruplex DNA” by Morel et al. submitted to Molecules.
The development of novel anticancer drugs having different biomolecular targets than known to date, is an interesting area in academia and pharma. In particular, G-quadruplex (G4) DNA has emerged as a potential molecular target for anticancer therapies because of its significance difference over the most common double stranded DNA. Novel approaches can overcome the resistance acquired of several cancers towards drugs in clinics such as cisplatin and in addition, lower the toxicity of current treatments. Having this in mind, Teulade-Fichou’s team describes the design, synthesis and characterization of new terpyridine-based metal complexes as molecules for G-quadruplexes. The portfolio of complexes has a platinum ion, which can coordinate DNA in a similar mechanism than cisplatin and thus, can potentially be used as a drug in anticancer therapies. Upon molecule-development, the interaction of the portfolio of complexes has been studied by a range of biophysical techniques such as FRET melting, FID and electrophoretic assays. They evaluated the interaction towards a panel of G4 DNAs of different topology (parallel, antiparallel and hybrid) to assess the topology preference, together with a double stranded model as a control to assess the selectivity of G4 vs. duplex. The covalent interaction to DNA has been extensively studied and well-analysed by electrophoresis. Finally, the cell viability of the complexes towards sensitive and resistant cisplatin tumour cell lines has been assessed. The insights enlightened in this work are of great importance and relevance for a large community of scientists, which can contribute to the rational design of novel coordinative G4 binders in anticancer therapies. To highlight that the work is well performed, analysed and discussed and altogether, this work constitutes a nice piece of work that shall be published in Molecules. Nevertheless, a few comments and suggestions can be useful to improve the quality of the work and clear some grey spots in my view.
(1) For the biophysical analysis of inorganic molecules, it is of special relevance to be confidence in the molecular weight of the complexes. A small variation of the molecular weight can influence the concentration of the stock solutions, which afterwards will be used for the studies. Frequently, metal complexes show a larger quantity of salts, which are not detected by NMR, mass spectrometry or HPLC chromatogram but can be assessed by elemental analysis. Could the authors provide some insights about this point since the complexes characterization only display the NMR and LC-MS?
(2) Figures 1 and 3 show a radar plots of the ΔTm or % probe displacement of the complexes but the selected colors difficult the reader to visualize, analyze and follow the discussion. Can the authors change these plots to get a better contrast among complexes and visualize much better? Either by change the colors, size, dots…
(3) The authors could introduce the reference of the first-in-class anticancer drug RSH4 as a proof-of-concept of G-quadruplex DNA molecules in anticancer therapies.
(4) Could the authors justify the use of 21CTA as an antiparallel G4? As far as it is described in the literature this G4 has a GCGC tetrad instead a G4 tetrad.
(5) Have the authors studied the stability of the complexes in biological media? An important factor linked to the biological activity can be the substitution of the chloride by an aqueous molecule or other anionic species and thus, the stability in media can give insights into this point and clarify the discrepancy between in vitro and cellular studies.
(6) Depending on the section, some words are different written such as c-myc*/ds26* (lines 235, 238, 240, 245, 247 and 248) but written without asterisk in the rest of the manuscript or CH2Cl2 (line 408 and DCM (line 420). Can the authors change them and homogenize the text?
(7) Some acronyms shall be changed such as 1H, 13C, (line 356) DMSO-d6 (line 358, 382, 412, 438…) for the correct spelling 1H, 13C, DMSO-d6
Author Response
Review of the manuscript Molecules 417755 entitled “Selectivity of terpyridine platinum anticancer
drugs for G-quadruplex DNA” by Morel et al. submitted to Molecules. The development of novel anticancer drugs having different biomolecular targets than known to date, is an interesting area in academia and pharma. In particular, G-quadruplex (G4) DNA has emerged as a potential molecular target for anticancer therapies because of its significance difference over the most common double stranded DNA. Novel approaches can overcome the resistance acquired of several cancers towards drugs in clinics such as cisplatin and in addition, lower the toxicity of current treatments. Having this in mind, Teulade-Fichou’s team describes the design, synthesis and characterization of new terpyridine-based metal complexes as molecules for G-quadruplexes. The portfolio of complexes has a platinum ion, which can coordinate DNA in a similar mechanism than cisplatin and thus, can potentially be used as a drug in anticancer therapies. Upon molecule-development, the interaction of the portfolio of complexes has been studied by a range of biophysical techniques such as FRET melting, FID and electrophoretic assays. They evaluated the interaction towards a panel of G4 DNAs of different topology (parallel, antiparallel and hybrid) to assess the topology preference, together with a double stranded model as a control to assess the selectivity of G4 vs. duplex. The covalent interaction to DNA has been extensively studied and well-analysed by electrophoresis. Finally, the cell viability of the complexes towards sensitive and resistant cisplatin tumour cell lines has been assessed. The insights enlightened in this work are of great importance and relevance for a large community of scientists, which can contribute to the rational design of novel coordinative G4 binders in anticancer therapies. To highlight that the work is well performed, analysed and discussed and altogether, this work constitutes a nice piece of work that shall be published in Molecules. Nevertheless, a few comments and suggestions can be useful to improve the quality of the work and clear some grey spots in my view.
We thank the Reviewer for his comments and recommendations, and we revised the manuscript accordingly. The modifications are boxed in yellow in the revised manuscript.
(1) For the biophysical analysis of inorganic molecules, it is of special relevance to be confidence in the molecular weight of the complexes. A small variation of the molecular weight can influence the concentration of the stock solutions, which afterwards will be used for the studies. Frequently, metal complexes show a larger quantity of salts, which are not detected by NMR, mass spectrometry or HPLC chromatogram but can be assessed by elemental analysis. Could the authors provide some insights about this point since the complexes characterization only display the NMR and LC-MS?
Metal complexes used in the present study are usually prepared from Cl2Pt(COD) or Cl2Pt(DMSO)2 . In the first case no salts are generated whereas KCl is formed in the second case but thorough washing with water removes the excess of salts. Finally all Terpyridine metal complexes are washed extensively with DCM and methanol /DCM that enables removal of the unreacted starting complexes (detectable by NMR). Therefore we are confident in the degree of purity of our complexes that is estimated around 90%.
(2) Figures 1 and 3 show a radar plots of the ΔTm or % probe displacement of the complexes but the selected colors difficult the reader to visualize, analyze and follow the discussion. Can the authors change these plots to get a better contrast among complexes and visualize much better? Either by change the colors, size, dots…
The modifications have been done.
(3) The authors could introduce the reference of the first-in-class anticancer drug RSH4 as a proof-of-concept of G-quadruplex DNA molecules in anticancer therapies.
It has been introduced
As an example, RHPS4 was one the first drug proof-of-concept of G-quadruplex DNA molecules in anticancer therapies [30]
(4) Could the authors justify the use of 21CTA as an antiparallel G4? As far as it is described in the literature this G4 has a GCGC tetrad instead a G4 tetrad.
The basic idea was to determine if some complexes show topology preference for certain G4 structures .Therefore a classical panel of G4 forming sequences with different toplogies was used. In this regard 21 CTA is one of the rare sequences that has been shown to adopt a well-defined antiparallel structure solved by NMR [Lim, K. W. et al. Sequence variant (CTAGGG)n in the human telomere favors a G-quadruplex structure containing a G·C·G·C tetrad. Nucleic Acids Res. 37, 6239–6248 (2009)].
(5) Have the authors studied the stability of the complexes in biological media? An important factor linked to the biological activity can be the substitution of the chloride by an aqueous molecule or other anionic species and thus, the stability in media can give insights into this point and clarify the discrepancy between in vitro and cellular studies.
The ligand exchange of Pt-ttpy in presence of G4 DNA in physiological buffer has been previously studied by us using ESI-MS which was reported in a former study [E.Largy et al. Chem.Eur.J. 2011,7, 13274]. Stability of the new complexes described herein might be similar but this has not been checked systematically. However this is an interesting suggestion and this aspect will be investigated in the future to understand better the biological properties and eventually rationalize discrepancy between in vitro and cellular studies. However, the cytotoxicity of the complexes might not be exclusively linked to the capacity to coordinate DNA as we observed with Pt(PA)-tpy, that is deprived of labile ligand and exhibits a good cytotoxicity.
(6) Depending on the section, some words are different written such as c-myc*/ds26* (lines 235, 238, 240, 245, 247 and 248) but written without asterisk in the rest of the manuscript or CH2Cl2 (line 408 and DCM (line 420). Can the authors change them and homogenize the text?
c-myc* or ds26*is used for radiolabeled oligonucleotides
(7) Some acronyms shall be changed such as 1H, 13C, (line 356) DMSO-d6 (line 358, 382,
412, 438…) for the correct spelling 1H, 13C, DMSO-d6
Round 2
Reviewer 1 Report
In the revised version, E. Morel et al. have essentially addressed all the major concerns raised by the
reviewers.
In my opinion, the article has been sensibly improved and can now be accepted for publication in Molecules.
Only few minor revisions are necessary, listed below.
1) In the Abstract, the authors should mention that the studied Pt(II)-complexes have been tested also
on one normal cell line.
2) Pag. 1, l.35: “the latter being form by four guanines” should be “the latter being formed by four
guanines”;
3) Pag. 2, l.45: “for the application of G-quadruplex DNA molecules in anticancer therapies” should be
“for the application of G-quadruplex-binding molecules in anticancer therapies”;
4) Pag. 2, l.53: “others complexes were shown to target” should be “other complexes were shown to
target”;
5) Pag. 9, l. 245: “given rise to many platinated products” should be “giving rise to many platinated
products”;
6) Pag. 9, l. 260: “The selectivity of c-myc platination produces by Pt-ttpy and Pt-tpy” should be “The
selectivity of c-myc platination produced by Pt-ttpy and Pt-tpy”;
7) Pag. 10, ll. 273 and 274: “platination kinetic are affected” should be “platination kinetics are
affected”;
8) Pag. 10, l. 278: “These results corroborate with the low binding selectivity” should be “These results
confirm the low binding selectivity”;
9) Pag. 11, l.337: “To be underlined” could be better replaced by “Noteworthy”;
10) Pag. 11, l.346: “if compare to” should be “if compared to”;
11) Pag. 12, l.348: “slow exchange kinetic of the chloride” should be “slow exchange kinetics of the
chloride”;
12) ll. 670 and 732: correct the numbers of the references, 4 should be 24 and 5 should be 45.
Author Response
We thank the reviewer for the English spelling corrections that has been done in the final manuscript. The modifications are in yellow